# Characterisation of the Contact between Cross-Country Skis and Snow: A Micro-Scale Study Considering the Ski-Base Texture

Kalle Kalliorinne [1,*], Bo N. J. Persson [2,3], Joakim Sandberg [1], Gustav Hindér [1], Roland Larsson [1], Hans-Christer Holmberg [4] and Andreas Almqvist [1]

[1] Division of Machine Elements, Luleå University of Technology, 971 87 Luleå, Sweden; joakim.sandberg@ltu.se (J.S.); gustav.hinder@ltu.se (G.H.); roland.larsson@ltu.se (R.L.); andreas.almqvist@ltu.se (A.A.)

[2] Peter Grünberg Institute (PGI-1), Forschungszentrum Jülich, 52428 Jülich, Germany; b.persson@fz-juelich.de

[3] Multiscale Consulting, 52428 Jülich, Germany

[4] Division of Health, Medicine and Rehabilitation, Luleå University of Technology, 971 87 Luleå, Sweden; integrativephysiobiomech@gmail.com

[*] Correspondence: kalle.kalliorinne@ltu.se

**Abstract:** In winter sports, the equipment often comes into contact with snow or ice, and this contact generates a force that resists motion. In some sports, such as cross-country skiing, this resistive force can significantly affect the outcome of a race, as a small reduction in this force can give an athlete an advantage. Researchers have examined the contact between skis and snow in detail, and to fully understand this friction, the entire ski must be studied at various scales. At the macro scale, the entire geometry of the ski is considered and the apparent contact between the ski and the snow is considered and at the micro-scale the contact between the snow and the ski-base textures. In the present work, a method for characterising the contact between the ski-base texture and virtual snow will be presented. Six different ski-base textures will be considered. Five of them are stone-ground ski bases, and three of them have longitudinal linear textures with a varying number of lines and peak-to-valley heights, and the other two are factory-ground "universal" ski bases. The sixth ski base has been fabricated by a steel-scraping procedure. In general, the results show that a ski base texture with a higher $S_{pk}$ value has less real contact area, and that the mutual differences can be large for surfaces with similar $S_a$ values. The average interfacial separation is, in general, correlated with the $S_a$ value, where a "rougher" surface exhibits a larger average interfacial separation. The results for the reciprocal average interfacial separation, which is related to the Couette type of viscous friction, were in line with the general consensus that a "rougher" texture performs better at high speed than a "smoother" one, and it was found that a texture with high $S_a$ and $S_{pk}$ values resulted in a low reciprocal average interfacial separation and consequently low viscous friction. The reciprocal average interfacial separation was found to increase with increasing real contact area, indicating a correlation between the real area of contact and the Couette part of the viscous friction.

**Keywords:** winter sports; sports equipment; snow; cross-country skiing; ski friction; ski-base texture

## 1. Introduction

In the contact between the equipment utilised in most winter sports and the snow/ice, kinetic energy is dissipated into heat due to friction. In some cases, friction is the main contributing force resisting the athlete's forward movement [1]. On such an occasion, a relatively small reduction in the frictional losses associated with cross-country skiing can allow greater speed or reduce the energy consumption on certain segments of the track, sometimes thereby determining the outcome of a race [2].

As early as 1939, Bowden and Hughes [3] reported in detail on the influence of the load, materials involved, and temperature on contact friction with snow. As is now well-known,

their findings revealed that the kinetic friction is lower than the static friction, and that below $-3\,^\circ\text{C}$ (the temperature of the ice cave where they performed their measurements never rose above $-3\,^\circ\text{C}$), the friction increases with decreasing temperature. A decade later, Eriksson [4] presented several hypotheses concerning how different snow conditions and surface textures influence friction. Some years later, Bowden [5] observed that a ski base consisting of polytetrafluorethylene has lower friction, setting the stage for the modern use of fluor-based ski bases and waxes (which are now partly forbidden in official competitions, because of their environmental hazard).

Researchers' interest in friction on snow remains high and many new theories for what is causing it have been put forth. For example, Lever and colleagues [6] describe a hypothesis that a thin quasi-liquid layer on snow crystals is prone to shear, but may be too thin to separate surfaces with micro-scale roughness that can carry the load and, therefore, that abrasion of snow particles may govern friction.

There are numerous models designed to predict friction on snow and ice that use the apparent area of contact, although more refined estimates of the actual area of contact are considered as well. For example, both Glenne [7] and Makkonen [8] incorporated expressions for the *dry* contact area ($A_{dry}$) based on the idea that the stresses in the contact between the ski base and the snow everywhere equal the unconfined compressive strength of the snow ($\sigma_{ucs}$), i.e., $A_{dry} = P/\sigma_{ucs}$, where $P$ is the total load. However, this does not take into account that ski bases made of different materials and emerging from different fabrication procedures (see Aghababei et al. [9]), will contact the snow in different ways, and therefore that the contact pressure, in reality, is not equal to the unconfined compressive strength of the snow everywhere.

Other models [10–12], including the authors' previous work [13,14], have only considered the apparent area of contact, an approach that completely neglects the influence of the preparation of the ski base. Recently, Lever and colleagues [15] utilised a setup involving a rotating disc to observe the thermal and mechanical characteristics of polyethylene in sliding contact with snow. These investigators observed that the contact area increases with sliding distance, encompassing almost 30% of the base after sliding for several hundred meters.

The texture of the ski base has, not surprisingly, been shown to exert a considerable impact on ski–snow friction, see, e.g., [16]. In the case of miniature skis, Giesbrecht and colleagues [17] found that an optimal $R_a$ value for the texture of the ski base was in the range of 0.5–1 µm. When Rohm and co-workers [18] compared two ski bases with completely different textures, they found that they performed similarly at $-11.1\,^\circ\text{C}$, but one performed better at lower and the other at higher temperatures. They hypothesised that this could be related to the surface's ability to form contact area. They finally concluded that the friction between the ski base and the snow cannot be characterised by average roughness parameters (e.g., $S_a$ and $S_z$), suggesting that the characterisation also needs to be based on (a set of) functional parameters related to the tribology of the ski base–snow contact, see, e.g., Persson [19].

Although examination of contact mechanics on the micro-scale has been part of standard procedure in connection with tribological evaluation for some time, in the case of cross-country skiing there are only a few such publications regarding the contact between the ski base and snow. One example is the work by Bäurle and his colleagues [20], who examined the micro-scale contact properties of cross-country skis utilising X-ray micro-computed tomography and numerical simulations, revealing a relative real area of contact of 6.4% at an apparent pressure of 30 kPa. These investigators also estimated that melting might increase the relative real area of contact to 25% and, under warm conditions, even to as much as 100%. Another study is the work by Scherge et al. [16], where they employed an already established numerical contact mechanics approach to quantify the contact area.

Theile and co-workers [21] also employed X-ray micro-computed tomography, and they concluded that most of the load on cross-country skis was concentrated on a small region of the total area, with an actual contact area of 0.4%. Before and after 50 trials at $-2$ and $-18.5\,^\circ\text{C}$, Rohm and colleagues [22] analysed both the effect of wear on ski waxes and

of the snow porosity. At both temperatures, the initial porosity of the snow at a depth of 0.1 mm was 70%, but after 50 runs at $-2\,^{\circ}$C the snow had been compacted to a porosity of 20%, whereas at $-18.5\,^{\circ}$C the snow had undergone less compaction, resulting in 45% porosity. Recently, applying X-ray micro-computed tomography as well, Mössner and co-workers [23] simulated the contact between a single grain of snow and the ski base and found a relative real area contact as high as 3%. They also observed that the porosity of their snow was 79%.

All in all, there is as yet no reliable and effective method for determining the real area of contact between a ski base and snow. In addition, the snow surface's porosity has not been considered in simulations. Accordingly, here we present an approach to determining the real contact area and the average interfacial separation, as well as other important contact parameters that characterise the contact between ski bases, with various textures and snow of different surface porosities.

## 2. Theory

While the ultimate goal is to simulate the contact between the ski base and the track, from tip to tail, with high enough resolution to resolve the ski-base texture and an arbitrary degree of porosity of the surface of the snow in the ski track, this is not yet feasible. We have, therefore, chosen to adopt a multi-scale framework. In fact, our intention is to establish a two-scale model by combining a model governing the macro-scale [14] with a micro-scale model. In this section, we present the theory and the assumptions for the approach employed for estimating the real area of contact, the average interfacial separation, corresponding to the volume of the void space between the ski-base surface and the snow, and the average reciprocal interfacial separation, which all are related to the micro-scale of the problem. Note that the present analysis pertains to the solid–solid contact, but the problem also involves solid–liquid contact (between the ski base and the snow), i.e., in the hydrodynamically pressurised water film, and in the water film and/or bridges in the surrounding areas. Figure 1 gives a graphical illustration of the methodology used in the numerical analysis of the micro-scale problem.

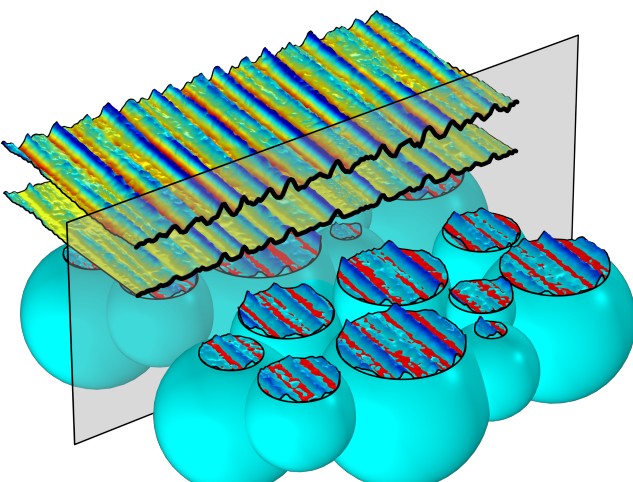

**Figure 1.** Illustration of the methodology used in the numerical analysis. The top part shows an inverted colour map of a measured ski-base texture from a cross-country ski, with the contacting surface facing downwards. The part below displays the measured surface when pressed against a block of ice with a perfectly smooth surface (not visible). The bottom part intends to illustrate the porosity of the snow, with a number of *worn* snow grains (illustrated as spherical particles with shaved-off upper parts that are all levelled to the same plane) which are in contact with the ski base. The red and blue colour map is of the ski-base topography (which is in contact with the worn snow grains); it indicates contact with red and the interfacial separation in shades of blue–the darker the blue, the larger the separation.

The top part shows a coloured height map of the (inverted) topography of a measured ski-base texture from a cross-country ski. The part below displays the topography of the measured surface when pressed against a block of ice with a perfectly smooth surface (not visible). The bottom part intends to illustrate the surface porosity of the snow with a number of *worn* snow grains, as spherical particles with shaved-off upper parts (all levelled to the same plane), which are in contact with the ski base. It should, however, be emphasised that the snow grains are only there for illustrative purposes and that the present model of the porosity (defined in (1)) controls the load-bearing area of the virtual snow surface. The coloured height map (in red and shades of blue) applied to the ski base topography, which is in contact with the worn snow grains, indicates contact with the snow in red, and the interfacial separation in shades of blue–the darker the blue, the larger the interfacial separation. In the present paper, the focus will be on how to characterise ski-base structures based on six different surface roughness parameters, as well as on the real area of contact $A_r$, average interfacial separation $\bar{h}$, and average reciprocal interfacial separation $\overline{1/h}$. The average interfacial separation is defined as the mean vacant volume between the snow and the ski base, and the average reciprocal interfacial separation is a determining factor appearing in the expression for the Couette part of the viscous friction. The real area of contact will be presented in terms of the percentage of contact area relative to the total area $A_t$, i.e., $A_r/A_t \times 100$.

The main difference between snow and ice is their composition and structure. More precisely, snow is a type of granular material created through precipitation, while ice is considered to be a homogeneous, crystalline, solid structure, but both are formed when water freezes. In this work, snow is modelled as a porous material with the same mechanical properties as ice, and we consider the porosity as a parameter that distinguishes the porous snow surface from the ice. To this end, we define a parameter as the ratio between the pore surface area ($A_p$) (void surface area) and total surface area ($A_t$), i.e.,

$$n = \frac{A_p}{A_t}. \tag{1}$$

To estimate the contact mechanical response between a ski-base texture and a porous surface, a few assumptions have to be made. In the present micro-scale model, the nominal load is defined as the applied load distributed over the nominal contact area ($A_t$) of a nonporous surface ($n = 0$), and it is considered as a function of the displacement $\delta$, i.e., $p = p(\delta)$. Hence, when a surface porosity $0 \leq n \leq 1$ is introduced, the apparent pressure, i.e.,

$$p_n(\delta) = p(\delta) \cdot (1 - n), \tag{2}$$

is proportional to the nominal load ($p(\delta)$) and decreases linearly as the porosity ($n$) increases. This means that also the real area of contact, corresponding to the apparent pressure $p_n(\delta)$, will vary linearly with the porosity. That is,

$$A_{r,n}(\delta) = A_r(\delta) \cdot (1 - n), \tag{3}$$

where $A_r(\delta)$ is the real area of contact for a nonporous surface at the simulated nominal load $p(\delta)$, and $A_{r,n}(\delta)$ is the real area of contact, corresponding to the apparent pressure $p_n(\delta)$ at the degree of porosity introduced. This means that to obtain the contact parameters at a certain apparent load $p_n(\delta)$ and porosity $n$, the contact mechanical response has to be calculated at $p(\delta)/(1 - n)$ to accommodate for the reduction in apparent pressure due to porosity. The average interfacial separation $\bar{h}(\delta)$ will vary with the degree of porosity as a result of applying the apparent pressure $p_n(\delta)$, but it does not, however, scale linearly with the apparent pressure $p_n(\delta)$ and contact area $A_{r,n}(\delta)$, hence

$$\overline{h}_n(\delta) = \overline{h}(\delta) = \frac{1}{\|\Omega\|} \int_\Omega h(\delta)\, dS, \tag{4}$$

where $\Omega$ is the computational domain, with area $\|\Omega\| = A_t$.

The average of the reciprocal interfacial separation is an important parameter when characterising the Couette part of the viscous friction, which originates in the hydrodynamically pressurised water film, and in the water film in the surrounding areas. Here, only solid–solid contact is considered. Thereby, the average of the reciprocal interfacial separation is considered in the areas surrounding the solid–solid contacts. Since the pore area does not contribute here, it is assumed that the average reciprocal interfacial separation also will vary linearly with the degree of porosity (in the same way as the apparent pressure and the real area of contact), and it is defined as

$$\overline{1/h(\delta)}_n = \frac{1-n}{\|\Omega_g\|} \int_{\Omega_g} \frac{1}{h(\delta)}\, dS, \tag{5}$$

where $\Omega_g = \Omega \backslash \Omega_c$ is the part of the domain where there is a gap (and possibly solid–liquid contact) between the surfaces and $\Omega_c$ is the part of the domain where there is solid–solid contact.

### 3. Method

The characterisation procedure employed in this work to calculate the six surface-roughness parameters listed in Table 1, to estimate the real area of contact ($A_{r,n}$), the average interfacial separation ($\overline{h}_n$) (volume of the void space between the ski-base surface and the snow), the average of the reciprocal interfacial separation ($\overline{1/h}_n$), and the apparent contact pressure $p_n$, consists of three steps: (1) surface topography measurement calculation of the roughness parameters, (2) contact mechanics simulation, and (3) analysis. These steps are schematically illustrated in Figure 2.

**Table 1.** Surface roughness parameters for the ski-base textures shown in Figure 4.

| Textures | $S_a$ (µm) | $S_q$ (µm) | $S_{sk}$ (-) | $S_{ku}$ (-) | $S_{dq}$ (µm/mm) | $S_{pk}$ (µm) | $S_k$ (µm) | $S_{vk}$ (µm) |
|---|---|---|---|---|---|---|---|---|
| Linear 1 | 1.767 | 2.159 | −0.16 | 2.58 | 65.79 | 1.444 | 6.009 | 1.920 |
| Linear 2 | 2.437 | 3.011 | −0.49 | 2.89 | 72.70 | 1.571 | 7.837 | 3.546 |
| Linear 3 | 8.768 | 9.622 | −0.43 | 1.63 | 185.77 | 2.161 | 11.651 | 20.019 |
| Brand A | 4.999 | 6.305 | −1.13 | 3.49 | 168.46 | 1.917 | 9.416 | 13.196 |
| Brand B | 4.881 | 5.803 | −0.58 | 2.38 | 112.82 | 1.217 | 12.695 | 7.965 |
| Steel | 1.693 | 2.126 | −0.09 | 3.34 | 94.25 | 1.925 | 5.499 | 2.149 |

In the first step, the topography of the ski-base structure is measured directly or indirectly by using a replica of the ski base. The main advantage of using a replica of the ski-base surface instead of the ski base itself is the possibility of collecting samples during testing, eliminating the need to take the skis to the lab, see Jolivet et al. [24] for related work using this principle. In the present work, a ZYGO NewView 9000 white-light interferometer was employed to measure the topographies of six different ski-base structures (fabricated by either stone-grinding or steel-scraping) from replicas, and thereafter the surface roughness parameters were calculated. The measurements were conducted using an objective with $2.75\times$ magnification and a $0.5\times$ FOV lens. The measurement resolution was 6.3 µm and the area was $6.3 \times 6.3$ mm$^2$ and a low-pass filter was applied to remove wavelengths shorter than 50 µm. The height probability density (HPD) and power spectra (PSD) of the six ski-base textures are presented in Figure 3. Both the HPD and PSD display valuable information about the surfaces, which is particularly useful when analysing the functional properties of the surfaces.

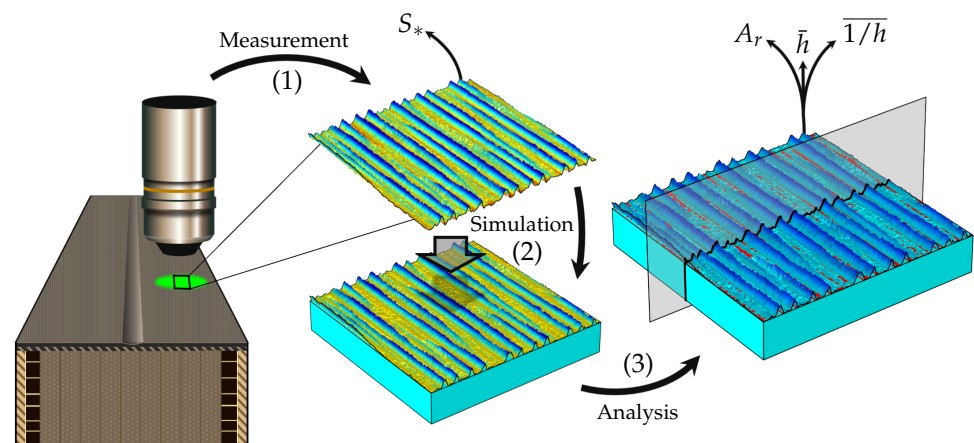

**Figure 2.** Illustration of the three-step characterisation procedure used in this work. (1) A measurement is taken of the ski-base structure and the seven surface roughness parameters listed in Table 1 are calculated. (2) The contact between the inverted replica's measured surface topography and an ice counter-surface is simulated for a range of nominal loads (0–100 kPa). (3) The in-contact topographies are analysed at different loads, here a large number of parameters can be retrieved, e.g., $A_{r,n}$, $\bar{h}_n$, and $\overline{1/h_n}$, but the figure here depicts $A_{r,n}$ in red and $h$ in shades of blue, the darker the blue, the larger the value of $h$, for a given porosity $n$.

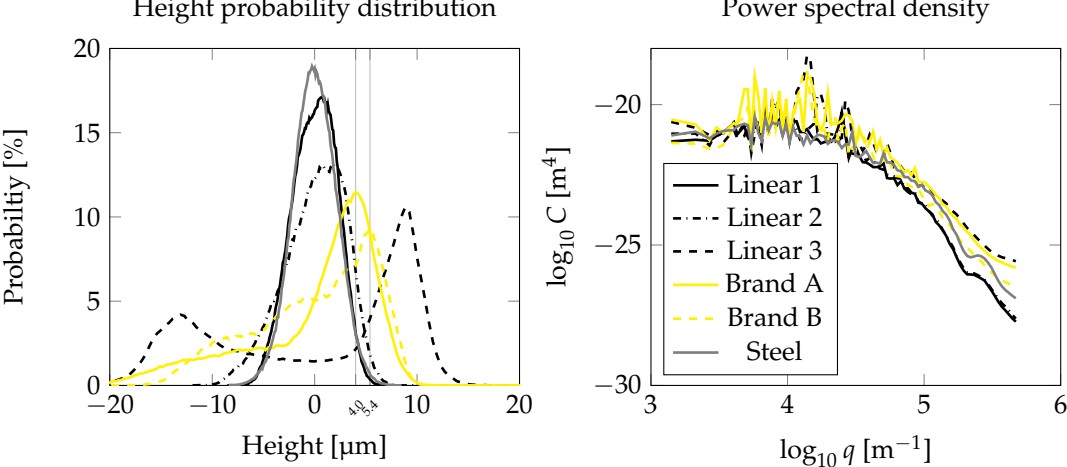

**Figure 3.** The height probability distributions (HPD) and power spectral densities (PSD) for the six surfaces. The left part shows the HPD, and it is interesting to note that the Linear 1 and 2 textures exhibit similar shapes but that Linear 3 stands out with the additional probability peak for the material in the bottom of its wide and deep grooves. It is also interesting to see the similarities and differences between the Brand A and Brand B textures. Both have a similar load-bearing material ratio (evidenced by the peaks at ≈4 and ≈5.4 µm, respectively, but with rather different HPD at lower heights). The similarity between the seemingly Gaussian steel-scraped texture and Linear 1 (the latter with a more blunt peak than the former) is also noted. The right part shows the PSD (in log–log scale), and the green dashed line with slope $-4$ and Hurst exponent $\mathcal{H} = 1$ has been included to give an idea of the influence that the high-frequency content might have on the contact mechanics behaviour.

In the second step, a simulation of the contact between the inverted replica's measured surface topography and an ice counter-surface, for a range of nominal loads, was performed. To this end, the boundary-element-based method (BEM), developed by Almqvist et al. [25], and later improved by Sahlin et al. [26], was employed. We note that, since then, this type of model has been employed in several works, e.g., [27–30]. For the simulation, the surfaces are considered to be perfectly elastic and the material properties were specified using Young's moduli $E_{ice} = 9\,\text{GPa}$ and $E_{base} = 0.9\,\text{GPa}$, but with the same Poisson's ratio,

$\nu = 0.3$. We note that (in reality) both the ski base and the snow are prone to plastic and/or visco-elastic behaviours and that although the mean contact pressure ($\leq$100 kPa) does not exceed the penetration hardness of ice ($\approx$15–20 MPa), the apparent pressure might exceed the hardness at some locations within the real contact area. In Figure 2 the same colour map and colour range are used for the topography of the measured ski-base structure (1), and the corresponding deformed *in-contact* topography (2), to provide a clearer visualisation of the contact mechanical interaction between the ski base and the snow (3).

In the third and last step, an analysis of the contact between the ski-base structure and the ice surface is carried out. There are a large number of parameters that can be extracted here, e.g., $A_{r,n}$, $\overline{h}_n$, and $\overline{1/h_n}$. The third analysis step is illustrated in Figure 2, with the real area of contact $A_{r,n}$ coloured red, and a coloured height map in shades of blue, the darker the blue, the larger the interfacial separation $h$.

## 4. Results and Discussion

The method described in the previous section was employed to characterise six different ski-base textures and evaluate their functionality. Figure 4a–c show the results for three stone-ground ski bases having *longitudinal* linear textures with a varying number of lines and peak-to-valley heights. Note that the stone that grinds the ski is textured by a dressing procedure, where a diamond tip is swept from one side of the rotating stone to the other. Hence, the texture will not be perfectly longitudinal. Instead, it will exhibit a small lay (compare with the thread in a screw), which will be reflected in the ski-base texture after grinding.

Figure 4d,e show the results for two different factory-ground ski-base textures, which are meant to be "universal", implying that they should provide satisfactory performance in many different conditions, and Figure 4f for a steel-scraped ski base (fabricated by using a scraper made of steel with a sharp edge which is repeatedly used to cut away a very thin layer of material from the ski base).

The top part (1) in each of the six sub-figures (Figure 4a–f), shows the inverted height data, i.e., $-h_{ij}$, $1 \leq i,j \leq 1000$, for the measured ski-base texture, where blue in the coloured height map is the bottom of the valleys in the longitudinal stone-ground or steel-scraped scratches, and red is the top of the ridges that first contact the snow. Note that the colour range is set for each sample individually because of the large differences in peak-to-valley heights for the different ski-base textures. The magnification is, however, identical for each of the samples, which makes it possible to compare the amplitude of the textures and the black solid lines of the corresponding cross-section profiles. This is also true for the middle part (2), which displays the topography of the measured surface when it is pressed against a block of ice with a perfectly smooth surface (not visible). The bottom part (3) also shows the same deformed topography as in (2), but with a colour map that indicates the contact area with red, and the interfacial separation in shades of blue – the darker the blue, the larger the interfacial separation.

The analysis that now follows is divided into three sections. The first section presents a characterisation based on a selection of *standardised* surface roughness parameters. In the second section, the results obtained from contact mechanical simulations are presented in terms of *functional* parameters. In the third section, the present results are compared and discussed in relation to previous results made available by other researchers.

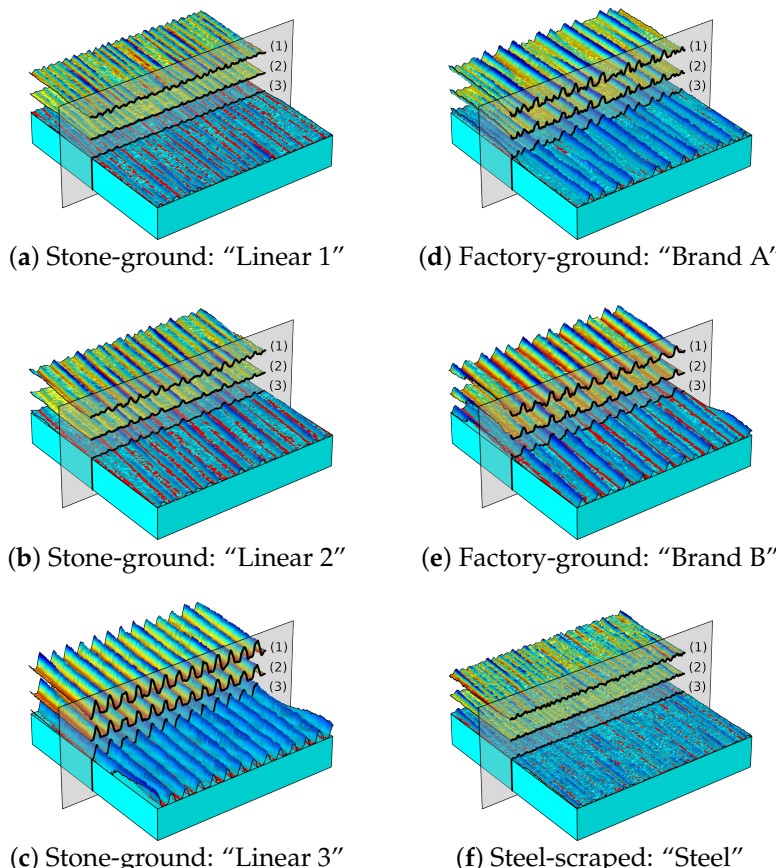

**(a)** Stone-ground: "Linear 1"

**(d)** Factory-ground: "Brand A"

**(b)** Stone-ground: "Linear 2"

**(e)** Factory-ground: "Brand B"

**(c)** Stone-ground: "Linear 3"

**(f)** Steel-scraped: "Steel"

**Figure 4.** Topographies of five ski bases produced by stone grinding and one by steel scraping. (1) The inverted height data for the measured ski-base texture, where blue in the colour map is the bottom of the valleys in the longitudinal stone-ground or steel-scraped scratches, and red is the top of the ridges that first contact the snow. (2) The *in-contact* topography of the measured surface (1), as pressed against a block of ice with a perfectly smooth surface (not visible). (3) The in-contact topography (2), but with a colour map indicating the contact area with red, and the interfacial separation in shades of blue—the darker the blue, the larger the interfacial separation. N.B. the same magnification is used for each sample, but the colour range is set for each sample individually, because of the large height differences in peak-to-valley heights for the different ski-base textures.

### 4.1. Standardised Surface Roughness Parameters

The values of seven surface roughness parameters for each of the six ski-base topographies are listed in Table 1. The $S_a$ value (which is the average of the absolute value of the difference between the surface's height data and its mean value, i.e.,

$$S_a = \frac{1}{N} \sum_{i,j}^{N} |h_{ij} - \bar{h}|,$$

where $N = 10^6$ is the total number of height data points) is often used as an estimate of how "rough" a surface is, and it is clear that the tabulated values are well correlated with the visual impressions conveyed by the figures. That is, based on the $S_a$ value, the Linear 3 topography is the "roughest", followed by the two factory-ground ski bases, Brand A and Brand B, which have similar $S_a$ values. Then comes the Linear 2 and the Linear 1 topographies, that exhibit an increasing number of lines and decreasing peak-to-valley heights compared to each other. The steel-scraped texture has the "smoothest" topography, with the $S_a$ value being approximately 19% and 34% of the Linear 3 and the two factory-ground ski bases, respectively.

The rms-roughness parameter $S_q$, which ranks the present surfaces in the same way as the $S_a$, is included for completeness. For a randomly rough surface with a Gaussian distribution $S_q = S_a\sqrt{\pi/2} \approx 1.25 S_a$. The HPD of the steel-scraped surface (Figure 3) indicates that it is close to Gaussian, and it does also have $S_q/S_a \approx 1.256$.

The skewness parameter, $S_{sk}$, is a measure related to the shape surface's height probability distribution, and a positive value indicates that the surface has more material below the core than above it and a negative value indicates the opposite. According to Table 1, the $S_{sk}$ values for all six of the surfaces considered are negative. This is also expected, since the topographies are fabricated by mechanical processes removing material under compressive loading conditions. It is, however, worth noting that the topography of the steel-scraped ski base has an $S_{sk}$ value close to zero, which is also an indicator that it might be Gaussian.

The kurtosis, $S_{ku}$, is often used in connection to the $S_{sk}$ value, and the higher the value is, the more "pointed" the surface's height probability distribution. It is also used in connection with characterisation by the $S_a$ value. For a Gaussian HPD, the higher the $S_{ku}$, the less the spread is, and for the steel-scraped ski-base texture the value is 3.34, indicating that it has a relatively narrow distribution (a stochastic variable with $S_{ku} > 3$ is said to have a *leptokurtic* distribution, and a randomly rough (Gaussian) surface has $S_{ku} = 3$). For surfaces that are not Gaussian, such as Brand A and Brand B, the larger value of the kurtosis of the former indicates that it has more localised peaks and valleys.

The root-mean-square slope parameter, $S_{dq}$, is a measure of how "sharp" the asperities are, as well as how "steep" the valleys are, and it can be used to distinguish between surfaces with similar $S_a$ values. Since the slope is related to the gradients of the height data it is very sensitive with respect to the quality of the measurement signal, and specifically the resolution in the height and lateral dimensions (in this case 55 nm and 6.3 μm, respectively). For this reason, it is also classified as a hybrid parameter, related to all three spatial dimensions. The $S_{dq}$ parameter can be useful for evaluating systems' sealing ability, assessing surface appearance, and determining the extent to which fluids wet a surface, i.e., the degree of hydrophobicity. As it is well-known that low wetting properties (high hydrophobicity) of the ski-base are essential while skiing in warmer conditions, it may be useful also when characterising ski-base surfaces, where a larger $S_{dq}$ value renders a higher degree of hydrophobicity [31,32].

For the three "linear" textures (Linear 1–3), it is clear that the coarser the longitudinal pattern, the higher the $S_{dq}$, and in this case it seems to be correlated with the $S_a$. However, while Brand A and Brand B have similar $S_a$ values, the difference in their $S_{dq}$ values is about 50%. The steel-scraped texture is the "smoothest" surface according to its $S_a$ value. The surface's root-mean-square slope is, however, higher than Linear 2 and not far from the significantly "rougher" Brand B texture.

Columns 4–6 in Table 1 declare the values of the reduced peak height $S_{pk}$, core roughness depth $S_k$, and the reduced valley depth $S_{vk}$ parameters, which are characteristics of the Abbott–Firestone curve [33,34]. The parameters are related to the surface's load-bearing capacity and the curve is often referred to as the bearing area curve. Due to the definition of these parameters, the sum of them, i.e., $\Sigma_k = S_{pk} + S_k + S_{vk}$, represents approximately the peak-to-valley height. The Linear 3 surface has the highest $S_{pk}$ (as well as the highest $\Sigma_k$). The steel-scraped (with the second lowest $\Sigma_k$) and the Brand A ($\Sigma_k$ second highest) surfaces do, however, have similar $S_{pk}$ values. Brand B is the surface with the lowest $S_{pk}$ value (but the third highest $\Sigma_k$), but it is similar to the values of the Linear 1 (with the lowest $\Sigma_k$) and 2 (with the third lowest $\Sigma_k$) surfaces.

### 4.2. Functional Parameters

Contact mechanics simulations of the six differently textured ski bases in contact with (virtual) snow were performed in order to study the variability in the percentage of the real area of contact (Figure 5), the average interfacial separation (Figure 6), and the average of the reciprocal interfacial separation (Figure 7), with respect to the apparent pressure ($p_n(\delta)$) and the porosity ($n$).

The results depicted in the left part of Figure 5, for a nonporous surface ($n = 0$), show that the contact area increases almost linearly with increasing load. According to contact mechanics theory [35], a linear relationship between the area of real contact and the nominal contact pressure up to ∼20% of complete contact is also predicted for randomly rough surfaces, which have a Gaussian height distribution. The results here show that this relation may be obeyed even for highly non-random surfaces such as the Linear 3 surface, which exhibits two peaks in the height distribution function (see Figure 3, left).

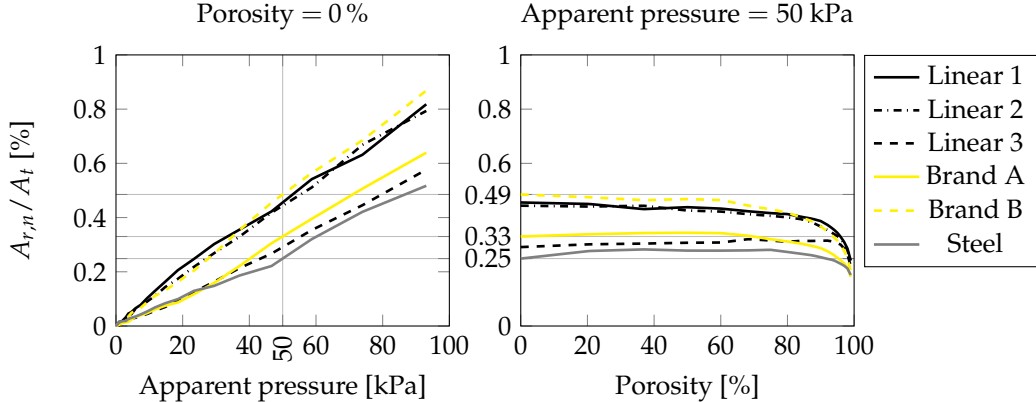

**Figure 5.** Relative real area of contact ($A_{r,n}/A_t$ in %) as a function of apparent pressure ($p(\delta)$) (**left**) and porosity ($n$) (**right**). The $A_{r,n}/A_t$ is indicated at 0% porosity and 50 kPa apparent pressure for the Brand A, Brand B, and steel-scraped surfaces.

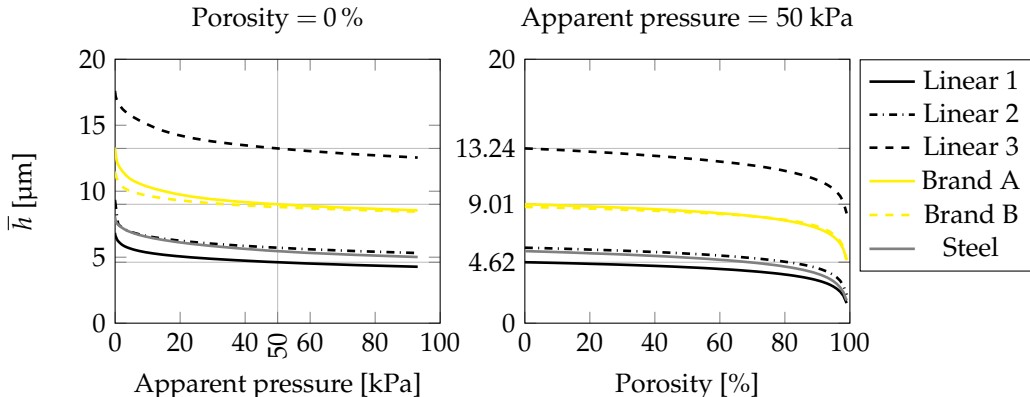

**Figure 6.** Average interfacial separation ($\bar{h}$) as a function of apparent pressure ($p_n(\delta)$) (**left**) and porosity ($n$) (**right**). The $\bar{h}$ is indicated at 0% porosity and 50 kPa apparent pressure for three of the surfaces.

Furthermore, the differences between the surface's contact areas (in general) increase with increasing load, and the surfaces cluster into two groups, i.e., (1) Linear 1, Linear 2, and Brand B, and (2) Linear 3, Brand A, and steel. The surfaces in Group 1 have lower $S_{pk}$ values than the surfaces in Group 2, and this seems to be correlated to the development of contact area with the load. More precisely, the results show that the surfaces in Group 2 with higher $S_{pk}$ develop less contact area when they carry the load than the surfaces in Group 1 do. It is, however, worth noting that, while Brand A develops the contact area faster than the Linear 3 ski-base texture, the latter has a higher $S_{pk}$ than the former. It should also be noted that the $S_a$ values for these three surfaces range from the lowest (steel-scraped) to the highest (Linear 3), suggesting that the $S_a$ value cannot be used as a determinant for the contact area.

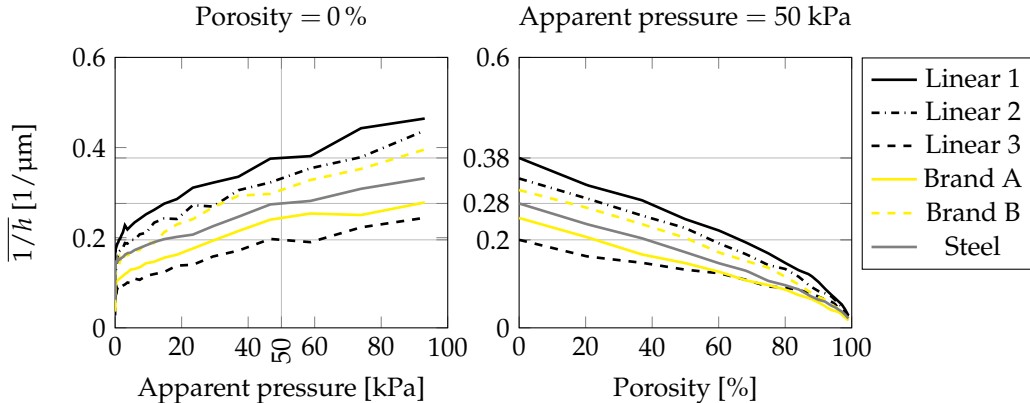

**Figure 7.** The average of the reciprocal interfacial separation ($\overline{1/h}$ in $1/\mu$m) as a function of apparent pressure ($p_n(\delta)$) (**left**) and porosity ($n$) (**right**). The $\overline{1/h}$ is indicated at 0% porosity and 50 kPa apparent pressure for three of the surfaces.

The right part of Figure 5 shows the resulting percentage of contact area obtained when varying the porosity, $n$, while keeping the apparent pressure constant at 50 kPa. It can be seen that an increase in the porosity (in general) results in a decrease in the contact area and that the ranking, established when varying the apparent pressure while keeping the porosity constant at 0%, is preserved for porosities $\lesssim$80%. As the porosity continues to increase, it can be observed that, (i) the Brand A, Brand B, and steel-scraped surfaces show a earlier decaying contact area than the three surfaces with linear textures, and (ii) the Linear 3 ski base retains an almost constant contact area $\approx$30% up to a much higher porosity ($\lesssim$95%) than the other surfaces.

A remark, in terms of the $S_a$ value, which might be used to characterise surfaces fabricated by the same manufacturing process, such as the three ski bases with linear textures in this case, it does not correlate well at all with the ranking in terms of the contact area, while the $S_{pk}$ value does. It should also be mentioned that, although there is a correlation between the peak area height parameter ($S_{pk}$) and the real contact area ($A_r$) for the surfaces studied here, it is the rms slope ($S_{dq}$) that determines $A_r$ for a randomly rough surface [19], while the $S_a$ (or rms-roughness ($S_q$)) amplitude is important for the average interfacial separation.

Another remark, in connection to the real area of contact and friction, is that making an effort to minimise the adhesive part, by decreasing the contact area, may result in an increase in the ploughing component of friction and possibly other parts as well. We note that the general consensus is that a "finer" texture has better performance in cold conditions. An example of such a surface building real area of contact efficiently with increasing load, is the Brand B texture having a high load-bearing area, i.e., significantly lower $S_{pk}$ value, but with a similar $S_a$ value as the Brand A texture.

Figure 6 shows the relation between the average interfacial separation and the nominal pressure (left) and the porosity (right). The ranking in terms of the average interfacial separation shown in the figure correlates well with the $S_a$ values listed in Table 1, i.e., the higher the $S_a$ value the higher the separation.

The average interfacial separation decreases at a higher rate at low apparent pressures and continues to decrease also at higher apparent pressures, but the differences between the surfaces are more or less constant for all apparent pressures ($p_n$) and porosities ($n$). The difference between the average interfacial separation for the Linear 1 and 3 surfaces is, for instance, approximately 300%. It seems like there is a correlation between a surface's $S_a$ value (and $S_q$) and $\overline{h}_n$, with the exception of the Linear 1 surface, which has a slightly larger $S_a$ but significantly lower $\overline{h}_n$ for all apparent pressures ($p_n$) and porosities ($n$).

Looking at the relation between average interfacial separation and porosity at an apparent pressure of 50 kPa, shown on the right of Figure 6, it is clear that an increase in porosity results in a decrease in the average interfacial separation. The average interfacial separation is linked to the texture's ability to accommodate excess water. Hence, a larger

interfacial separation helps to prevent the occurrence of a fully flooded water-lubricated condition and could, therefore, be beneficial in warmer conditions where there is more moisture in the snow and friction melting is more pronounced.

According to Persson's contact mechanics theory [36,37], for randomly rough surfaces the relation between the apparent (or nominal) pressure $p_n$ and the average interfacial separation is given by $\bar{h} = -b \ln(p_n/a)$, where $a$ and $b$ depends on the surface roughness power spectrum, and where $a$ is proportional to the effective elastic modulus $E^* = E/(1-\nu^2)$. In deriving this relation between $\bar{h}_n$ and $\ln p_n$ it is assumed that $A_{r,n}$ increases linearly with $p_n$, which is well obeyed in the present case (see Figure 5, left). The numerical results in Figure 6 (left) can be accurately fitted by this linear relation with the goodness of fit parameter $R^2$ ranging from 99.12% to 99.88%. For the steel-scraped surface, the height probability distribution is nearly Gaussian (see Figure 3, left) indicating a surface with nearly random (but anisotropic) roughness and, for this case, the theory predictions for $a$ and $b$ is in relatively good agreement with the fit parameters. In the past, the theory was only tested for isotropic roughness so this is the first test for anisotropic roughness.

Figure 7 depicts the average of the reciprocal interfacial separation ($\overline{1/h_n}$) as a function of apparent pressure (left) and porosity (right). Due to the definition of this parameter, it is connected to the Couette type of viscous friction induced by shearing the water film in the areas surrounding the solid–solid contact, and it is, therefore, of importance for this application. According to the general consensus, it is not possible to obtain full film lubrication while cross-country skiing. Actually, if it were, then the coefficient of friction, i.e., $\mu = \eta U A_{lub}(\overline{1/h})/(mg)$, where $\eta$ is the liquid's viscosity, $U$ the gliding speed of the ski, $A_{lub}$ the total lubricated area, and $m$ the body mass of the skier, would be much higher. For example, with $\eta = 1$ mPa s, $A_{lub} = 44 \times 200$ mm$^2$, $\overline{1/h} = 1/5$ μm$^{-1}$, $U = 10$ m s$^{-1}$, and $m = 30$ kg, then $\mu \approx 0.06$, which is larger than the friction coefficients typically observed in cross-country skiing, only in extreme situations would it be this large. Moreover, in the believed boundary/mixed lubrication situation, it is important to avoid excess water from coalescing, as it gives higher "capillary drag", caused by the increase in normal force due to capillary attraction.

Note that there will be two competing effects, i.e., the film thickness and the area covered by the meltwater. For rubber on a glass surface (e.g., wiper blades), the friction is maximal just before dry contact occurs due to water evaporation, see [38] for more about rubber friction. At the ski–snow interface the former will likely dominate, hence the strongest capillary effect is just when a meltwater film starts to appear. In this context, the topography should be such that it does not have a large $\overline{1/h_n}$, e.g., a topography providing mainly solid–solid contact regions with low shear resistance (and zero interfacial separation), surrounded by steep-walled grooves (with large interfacial separation) would be optimal.

The results presented in Figure 7 may, in general, be considered as the inverse of the results presented in Figure 6. It is, however, important to note that this is not literally the case. For example, there is a relatively large difference between Linear 3 and the steel surface when considering $\bar{h}_n$, while it is relatively small when considering $\overline{1/h_n}$. That is, although the Linear 3 surface has a significantly higher average interfacial separation than the steel-scraped surface, this may not affect the Couette part of the viscous friction that much (as the surfaces have similar $\overline{1/h_n}$). The general trend is that the group of three from Figure 5 is still visible, but it is reordered, where the highest $\bar{h}_n$ results in the lowest $\overline{1/h_n}$. In connection to the grouping of the surfaces with respect to the $S_{pk}$ parameter, it is clear that Group 2 (Linear 3, steel-scraped, and Brand A) with higher values, results in a lower $\overline{1/h_n}$ than the Group 1 (Linear 1, Linear 2, and Brand B) surfaces. Moreover, the results suggest that the ranking within the groups correlates to a combination of a high $S_{pk}$ and a high $S_a$ value that yields a low $\overline{1/h_n}$. More precisely, this is supported by the fact that the $S_{pk}$ values of the Group 1 surfaces and the $S_{pk}$ values of the Group 2 surfaces are similar, but the $S_a$ value (and $S_q$) is correlated with the $\overline{1/h_n}$ value.

Another result is that there is an approximately 90% difference between the Linear 1 texture and Linear 3 at a 50 kPa apparent pressure. This implies that the viscous friction, which is linearly dependent on the velocity ($U$) and the reciprocal average interfacial separation, i.e., $\propto U \cdot \overline{1/h_n}$, would be larger for all the other five surfaces considered here and, in particular, almost twice as large as for the Linear 3 surface. This is also in line with the general consensus that a "rougher" texture performs better at high speed than a "smoother" one (in terms of the $S_a$ value (or the $S_q$)), as long as they have similar $S_{pk}$. This reasoning holds for the whole range of apparent pressures ($p_n$) and porosities ($n$) lower than $\lesssim 80\%$.

Figure 7 (right) shows that $\overline{1/h_n}$ decreases with increasing porosity, but that the rate of change increases above approximately 70%. The Linear 1 surface, which shows the highest variability in $\overline{1/h_n}$ with $n$, also showed the highest variability in $A_n$ with $n$ (Figure 5). It is, however, the surface with the highest $\overline{1/h_n}$ at all apparent pressures ($p_n$) and porosities ($p_n$). On the contrary, although not providing the lowest $\overline{1/h_n}$ at zero porosity, the Brand A surface provides the lowest $\overline{1/h_n}$ at porosities higher than $\lesssim 80\%$. In other words, the results presented here indicate that in conditions where the surface porosity of the snow is higher, such as on newly groomed tracks, the glide of the steel-scraped surface may have a relative performance improvement compared with the other ski-base textures. In relation to this, the results also indicate that Linear 3 could perform better on polished (icy) ski tracks than the steel-scraped texture.

### 4.3. Comparison with Previous Results

In this section, we compare and discuss our findings with previous results made available by other researchers. More precisely, the results by Rohm et al. [18] and Scherge et al. [16].

#### 4.3.1. Rohm et al.

In Rohm et al. [18], they consider two ski-base textures with largely different HPD, see Figure 8. According to Rohm et al., the Ski 1 surface has more narrow grooves and ridges with broader plateaus than the Ski 2 surface, and based on their HPD, Ski 1 is referred to as the "bearing surface" and Ski 2 as the "nonbearing surface". By comparing the HPD of the Ski 1 surface and the six ski-base textures considered in the present work in Figure 3, it is most similar to the Brand A and Brand B surfaces. The "blunt" HPD of the Ski 2 surface (with an accumulation of material at $z \approx -3$ µm) does, however, differ quite substantially from the Ski 1 surface.

Table 2 lists four different surface roughness parameters for the Ski 1 and Ski 2 textures and it also includes rounded values for the six previously analysed ski-base surfaces to facilitate comparison and discussion. In addition, Table 3 presents reduced peak height ($R_{pk}$), core roughness height ($S_k$), valley depth ($S_{vk}$) *relative* to the peak-to-valley height, represented by $\Sigma_k = S_{pk} + S_k + S_{vk}$, the ratio $S_{vk}/S_k$, and $\Sigma_k$. According to the parameter values listed in the tables, the Ski 1 surface is most similar to the (factory-ground) Brand B surface, and Ski 2 stands out among all the surfaces in terms of its large $S_k/\Sigma_k$ and $(S_{pk} + S_k)/\Sigma_k$ values, and its small $S_{vk}/\Sigma_k$ and $S_{vk}/S_k$ values.

In relation to the results presented in Figure 5 for the variability in the percentage of real contact area (of the six ski-base textures considered in the present work), it was hypothesised that there might be a correlation between a low $S_{pk}$ value and a high $A_{r,n}/A_t$. From this hypothesis, it follows that (i) the Ski 1 ("bearing") surface would develop a contact area in parity with the Brand B surface, and (ii) the Ski 2 ("nonbearing") surface would build a real area of contact slower than all the other surfaces when increasing the apparent pressure. In connection to this, it was also hypothesised that a surface with both a high $S_{pk}$ value and $S_a$ value would exhibit a low average reciprocal interfacial separation $\overline{1/h_n}$. Hence, since the Ski 2 surface (with broad plateaus and narrow grooves), was found to perform better "on warm snow" in [18], it also ought to exhibit a lower $\overline{1/h_n}$ than Ski 1. In turn, this suggests it would be possible to use the present results (Figure 7) to discern

which ones, of the six ski-base textures characterised in this work, would perform well under warm conditions.

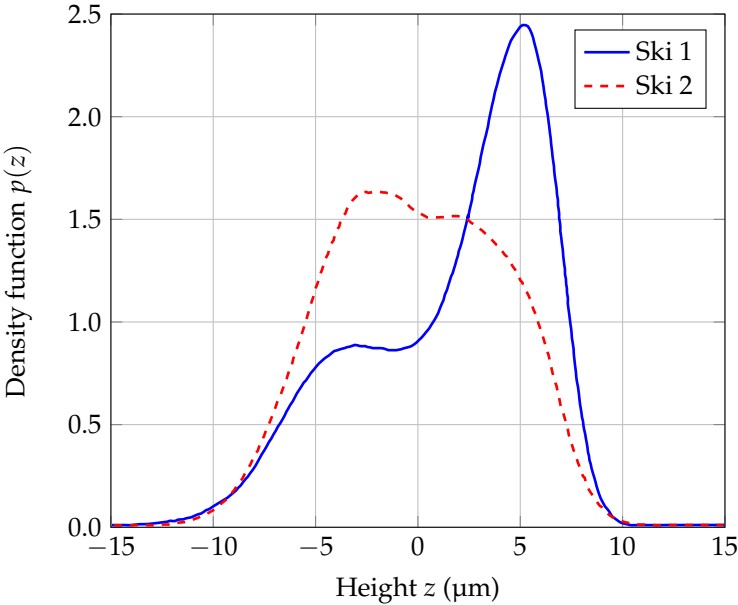

**Figure 8.** The height probability distribution of the "bearing surface" Ski 1 and the "nonbearing surface" Ski 2, adapted from Rohm et al. [18]. Reprinted with permission. Copyright © 2016 American Chemical Society.

**Table 2.** Surface roughness parameters for the two ski-base textures Ski 1 ("bearing surface") and Ski 2 ("nonbearing surface") considered in [18]. Rounded values for the six previously analysed ski-base surfaces have been appended to facilitate the comparison.

| Textures | $S_a$ (µm) | $S_{pk}$ (µm) | $S_k$ (µm) | $S_{vk}$ (µm) |
|---|---|---|---|---|
| Ski 1 | 3.60 | 1.40 | 9.00 | 6.55 |
| Ski 2 | 3.48 | 2.20 | 12.60 | 2.80 |
| Linear 1 | 1.77 | 1.44 | 6.01 | 1.92 |
| Linear 2 | 2.44 | 1.57 | 7.84 | 3.55 |
| Linear 3 | 8.77 | 2.16 | 11.65 | 20.02 |
| Brand A | 5.00 | 1.92 | 9.42 | 13.20 |
| Brand B | 4.88 | 1.22 | 12.70 | 7.97 |
| Steel | 1.69 | 1.93 | 5.50 | 2.15 |

**Table 3.** The reduced peak height ($R_{pk}$), core roughness height ($R_k$), valley depth ($R_{vk}$) *relative* to the peak-to-valley height, represented by $\Sigma_k = S_{pk} + S_k + S_{vk}$, the ratio $S_{vk}/S_k$, and $\Sigma_k$, for the two ski-base surfaces analysed in [18]. The corresponding values for the six previously analysed ski-base surfaces have been appended to facilitate the comparison.

| Textures | $S_{pk}/\Sigma_k$ | $S_k/\Sigma_k$ | $S_{vk}/\Sigma_k$ | $S_{vk}/S_k$ | $\Sigma_k$ (µm) |
|---|---|---|---|---|---|
| Ski 1 | 8.3% | 53.1% | 38.6% | 72.8% | 16.95 |
| Ski 2 | 12.5% | 71.6% | 15.9% | 22.2% | 17.60 |
| Linear 1 | 15.4% | 64.1% | 20.5% | 32.0% | 9.37 |
| Linear 2 | 12.1% | 60.5% | 27.4% | 45.2% | 12.95 |
| Linear 3 | 6.4% | 34.4% | 59.2% | 171.8% | 33.83 |
| Brand A | 7.8% | 38.4% | 53.8% | 140.1% | 24.53 |
| Brand B | 5.6% | 58.0% | 36.4% | 62.7% | 21.88 |
| Steel | 20.1% | 57.4% | 22.4% | 39.1% | 9.57 |

4.3.2. Scherge et al.

In [16], Scherge et al. presented results that were obtained "...under lab-conditions (Skitunnel Oberhof, Germany) with snow at a constant temperature of $-2\,°C$ and a humidity between 30 and 40%...". The results show that the real area of contact for five pairs of skis, with the different ski-base textures S1–5, is correlated with sliding time, which is related to friction. The surface roughness parameters for these ski-base textures are listed Table 4 together with the same set of surface roughness parameters corresponding to the six ski-base textures investigate here.

The ranking of the Group 1 surfaces (Linear 1, Linear 2, and Brand B), with higher $A_{r,n}/A_t$, and the Group 2 surfaces (Linear 3, steel, and Brand A), with lower $A_{r,n}/A_t$, can also be found for $\overline{1/h_n}$, even though there is no individual ranking within the groups. This indicates that there might be a correlation between the percentage of real area of contact and the average reciprocal interfacial separation, which is a determining factor in the Couette part of viscous friction. This can be attributed to the fact that as the contact area increases, the part $\Omega_g$ of the domain $\Omega$ where there is a gap between the surfaces decreases, and so does the interfacial separation. From (5), it is then clear that $\overline{1/h_n}$ increases. This suggests that the viscous friction increases with increasing total contact area, which may be the reason for the correlation observed in [16] between the real area of contact and the sliding time.

**Table 4.** Surface-roughness parameters for the ski-base textures considered in [16]. Rounded values for the six previously analysed ski-base surfaces have been appended to facilitate the comparison.

| Textures | $S_a$ (μm) | $S_{sk}$ (-) | $S_{ku}$ (-) | $S_{dq}$ (μm/mm) |
|---|---|---|---|---|
| S1 linear/fine | 1.86 | 0.41 | 1.68 | 135 |
| S2 linear/medium | 1.63 | 0.54 | 2.83 | 107 |
| S3 linear/coarse | 2.84 | 0.18 | 0.33 | 137 |
| S4 linear/mutliple | 2.45 | 0.29 | 0.70 | 175 |
| S5 cross-hatched | 1.81 | 0.67 | 2.22 | 126 |
| Linear 1 | 1.77 | −0.16 | 2.58 | 66 |
| Linear 2 | 2.44 | −0.49 | 2.89 | 73 |
| Linear 3 | 8.77 | −0.43 | 1.63 | 186 |
| Brand A | 5.00 | −1.13 | 3.49 | 168 |
| Brand B | 4.88 | −0.58 | 2.38 | 113 |
| Steel | 1.69 | −0.09 | 3.34 | 94 |

*4.4. Summary of the Results*

Regarding the real area of contact, it was found to increase approximately linearly with the apparent pressure and surfaces, with similar $S_{pk}$ values being grouped together. The group with higher $S_{pk}$ values developed less real area of contact area for porosities smaller than ($\lesssim$80%). Here, the nature of the variability changed and the real area of contact for the Brand A, Brand B, and steel-scraped surfaces showed a faster-decaying contact area than the three surfaces with linear textures. It was also observed that the Linear 3 texture retains an almost constant contact area of $\approx$30% up to a much higher porosity ($\lesssim$95%) than the other surfaces. The $S_a$ value, which might be used to characterise surfaces fabricated by the same manufacturing process, such as the three ski bases with linear textures in this case, did not correlate at all well with the ranking in terms of the contact area.

The average interfacial separation, which is related to the texture's capacity to accommodate excess water, showed a smoothly decreasing trend with both the apparent pressure and the porosity. The variability was, as expected, high in the ranges of low loads and high porosity. Surfaces with higher $S_a$ values (and $S_q$) showed larger average interfacial separations and, except for the Linear 1 and steel-scraped surfaces, the $S_a$ value could be used to predict the mutual order of the surfaces' average interfacial separations. For these two textures, with similar $S_a$ ($\approx$4.3% and even more similar $S_q$, i.e., $\approx$1.6%)), it was

the steel-scraped surface with the higher $S_{pk}$ value that resulted in the lowest average interfacial separation.

For surfaces with similar $S_{pk}$ values, the characterisation based on the average of the reciprocal interfacial separation, which is a functional parameter connected to the Couette type of viscous friction (that is linearly dependent on the velocity and the reciprocal average interfacial separation), was found to be in line with the general consensus that a "rougher" texture (high $S_a$ value) performs better at high speed than a "smoother" one (low $S_a$ value). This reasoning holds for the whole range of apparent pressures and porosities lower than ≲80%, suggesting that a surface with a high $S_{pk}$ value and a high $S_a$ value is beneficial for the viscous friction.

The ranking of the Group 1 surfaces (Linear 1, Linear 2, and Brand B), with a higher real area of contact, and the Group 2 surfaces (Linear 3, steel, and Brand A), with a lower real area of contact, can also be found for the reciprocal average interfacial separation, even though there is no individual ranking within the groups. This indicates a correlation between the real area of contact and the Couette part of viscous friction. This can be attributed to the fact that the reciprocal average interfacial separation increases as the contact area increases, suggesting that the viscous friction increases with increasing total contact area.

## 5. Conclusions

In the present study, a novel procedure for characterising the ski-base structure, involving three intermediate steps, was presented. The intermediate steps are: (1) high-quality optical surface topography measurement using replicas of the ski-base texture and calculation of standardised surface roughness parameters; (2) numerical simulation of the contact between the ski-base and a nominally flat surface, representing snow with different porosity; and (3) calculation and analysis of selected resulting functional parameters, i.e., the real area of contact, the average interfacial separation, and the average reciprocal interfacial separation.

Specific findings in the present work are:

- Surfaces with higher $S_{pk}$ values have lower contact area, but the $S_{pk}$ alone cannot be used to precisely predict the contact area.
- It was found that an increase in the porosity decreased the real area of contact, and ski-base textures with a larger real area of contact at $n = 0$ exhibited a higher variability.
- The surfaces were grouped by their $S_{pk}$ values and the group with the lower $S_{pk}$ values showed a higher rate of increase in contact area with increasing apparent pressure.
- The relative differences between the real area of contact for the Linear 3 ("roughest") and the steel-scraped surface ("smoothest"), and between the Linear 1 (second "smoothest") and the steel-scraped surface, at an apparent pressure of 50 kPa, were found to be ≈32% and ≈84%, respectively, indicating that the $S_a$ value is not correlated with the real area of contact.
- The differences between the average interfacial separation for the steel-scraped ("smoothest") and the Linear 3 surfaces ("roughest"), and the steel-scraped ("smoothest") and the Linear 1 surfaces (second "smoothest"), at a 50 kPa apparent pressure, were found to be ≈300% and ≈17%, respectively.
- The reciprocal average interfacial separation, hence the viscous part of the friction, is expected to be ≈50% higher for the Linear 1 than for the Linear 3 texture at a 50 kPa apparent pressure.
- The viscous friction is linearly dependent on the velocity and the reciprocal average interfacial separation ($\propto U \cdot \overline{1/h_n}$), and is larger for the Linear 1 texture than for all the other five surfaces considered here.
- The reciprocal average interfacial separation can be used to compare textures and possibly help to discern whether a texture performs well under warm conditions or not.

Finally, we remark that it has previously been concluded that one or a few standardised surface roughness parameters are insufficient to characterise the friction of different ski-base textures. We extend this by concluding that standardised surface roughness parameters are insufficient to quantitatively characterise the functional parameters considered herein and that it is yet to be validated whether these, possibly in combination with standardised surface roughness parameters, are correlated to the friction between the ski base and the snow.

**Author Contributions:** Conceptualisation, K.K., R.L. and A.A.; methodology, K.K. and B.N.J.P.; software, K.K.; validation, K.K.; formal analysis, K.K. and B.N.J.P.; investigation, K.K. and B.N.J.P.; resources, K.K., J.S. and G.H.; data curation, K.K.; writing—original draft preparation, K.K., J.S., G.H., R.L., H.-C.H. and A.A.; writing—review and editing, K.K., B.N.J.P., J.S., G.H., R.L., H.-C.H. and A.A.; visualisation, K.K., J.S. and G.H.; supervision, R.L., H.-C.H. and A.A.; project administration, A.A.; funding acquisition, R.L., H.-C.H. and A.A. All authors have read and agreed to the published version of the manuscript.

**Funding:** This research was funded by VR (The Swedish Research Council): DNR 2019-04293.

**Data Availability Statement:** The data presented in this study are available on request from the corresponding author.

**Acknowledgments:** The authors would like to acknowledge the support from SOK (Swedish Olympic Committee) and VR (The Swedish Research Council): DNR 2019-04293.

**Conflicts of Interest:** The authors declare no conflict of interest.

## Nomenclature

| | |
|---|---|
| $E_{ice}$ | Elastic modulus of ice $= 9$ GPa (Pa) |
| $E_{base}$ | Elastic modulus of the ski base $= 0.9$ GPa (Pa) |
| $\sigma_{ucs}$ | Unconfined compressive strength (Pa) |
| $\nu$ | Poisson ratio $= 0.3$ |
| $A_p$ | Pore surface area (m$^2$) |
| $A_t$ | Total surface area $A_t = \|\Omega\|$ (m$^2$) |
| $A_r$ | Real area of contact for a nonporous surface (m$^2$) |
| $A_{r,n}$ | Real area of contact for a porous surface (m$^2$) |
| $\Omega$ | Computational domain |
| $\Omega_c$ | The part of the domain where there is contact |
| $\Omega_g$ | The part of the domain where there is a gap (not contact) |
| $U$ | Sliding velocity (ms$^{-1}$) |
| $n$ | Surface porosity |
| $p$ | Nominal load (Pa) |
| $p_n$ | Apparent pressure (Pa) |
| $P$ | Load (N) |
| $\delta$ | Rigid body displacement (m) |
| $h$ | Interfacial separation (m) |
| $\bar{h}$ | Average interfacial separation (m) |
| $\overline{1/h}$ | Average reciprocal interfacial separation (m$^{-1}$) |

**Surface roughness parameters**

| | |
|---|---|
| $S_a$ | Arithmetic mean deviation |
| $S_q$ | Root-mean-square deviation |
| $S_{sk}$ | Skewness |
| $S_{ku}$ | Kurtosis |
| $S_{dq}$ | Root-mean-square slope |
| $S_{pk}$ | Reduced peak height |
| $S_k$ | Core roughness depth |
| $S_{vk}$ | Reduced valley height |

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
