# Peer review of "Characterisation of the Contact between Cross-Country Skis and Snow: A Micro-Scale Study Considering the Ski-Base Texture"

_lubricants, doi:10.3390/lubricants11050225_

Round 1

Reviewer 1 Report

In this manuscript, a method for characterising contact between ski-base texture and the snow is presented. Six ski-base textures were measured and the surface roughnesses were compared and analyzed. The real contact area was computed for the different apparent pressure and surfaces were obtained. The work is interesting and valuable. It is suggested to be accepted in the journal.

Author Response

See attached file for the author's response to all reviewers.

Reviewer 2 Report

In the present work, a method for characterising contact between the ski-base texture and virtual snow was presented. Six different ski-base textures has been considered, the real contact area and the interfacial separation were studied with different textures. The analysis on the surface roughness parameters were presented in detail, which provide valuable informations on describing surface. However, there are still some aspects need to be revised before the manuscript can be accept for publication.

1. When discussing the kurtosis Sku, that of Brand A is also larger than 3, which is similar with the Steel-scraped ski-base texture, further discussion on the this result is needed for better understanding.

2. The Sdq-parementer can be useful for evaluating the sealing ablity, how to use the Sdq to indicate the wetting properties of the ski base? For example, the Sdq of “Steel” sample is much lower than the Brand ones, what does it mean?

3. In the caption of Figure 5, the right figure shows the relative real are of contact as a function of prosity for six different surfaces, the writing in the caption is 3.

4. In the right part of Fig. 5, the fast decaying contact area also exhibited for the Linear 1 ski-base texture, Why?

5. The figure caption of Fig. 6 is wrong, it is the reciprocal interfacial separation rather than relative real area of contact.

6. In the conclusion part, the authors repeated the variation tendency in the results part, which is too long for the readers, please simplify the conclusions. 

The English writing is OK, however, there are some writting mistakes as suggested need to be revised.

Author Response

(The authors gave the same response as above.)

Reviewer 3 Report

The authors present a procedure for characterising and analysing the ski-base and snow interaction. In this procedure, optical surface topography measurements are taken to, later, numerically simulate the contact interaction between the ski-base and a nominally surface representing snow with different porosity. Finally, they compute and analyse: the resulting real contact area, the average interfacial separation, and the average reciprocal interfacial separation. The article is written in a comprehensible way and brings new knowledge about the ski-base and snow contact interaction problem. I have no major comments about the work..

Author Response

(The authors gave the same response as above.)
